# Female-Biased Expression of *R-spondin 1* in Chicken Embryonic Gonads Is Estrogen-Dependent

**DOI:** 10.3390/ani13132240

**Published:** 2023-07-07

**Authors:** Mingde Zheng, Xikui Liu, Yu Meng, Xiao Lin, Jiahui Li, Jianguo Zhu, Minmeng Zhao, Long Liu, Tuoyu Geng, Daoqing Gong, Jun Zhang

**Affiliations:** College of Animal Science and Technology, Yangzhou University, Yangzhou 225009, China; yzzheng410@163.com (M.Z.); 15227279522@163.com (X.L.); 15866226082@163.com (Y.M.); ydxiaolin@163.com (X.L.); lijiahui2994@163.com (J.L.); yzjianguozhu@163.com (J.Z.); zhaominmeng123@163.com (M.Z.); liulong@yzu.edu.cn (L.L.); tygeng@yzu.edu.cn (T.G.); yzgong@163.com (D.G.)

**Keywords:** chicken embryo, *RSPO1*, ovary, gonadal development

## Abstract

**Simple Summary:**

The *R-spondin 1* (*RSPO1*) gene is considered to play an important role in the gonadal development of female vertebrates, and estrogen plays an important regulatory role in the gonadal development of female chicken embryos. However, the expression of *RSPO1* in the gonad of chicken embryos and the relationship between *RSPO1* expression and estrogen level are not fully understood. In this study, we found that *RSPO1* showed female-biased expression in chicken embryonic gonads, but there was no significant difference in the expression of *RSPO1* between female and male gonadal cells. Moreover, the expression of *RSPO1* could be regulated by the alteration of estrogen levels at both cellular and in vivo levels, suggesting that the female-biased expression of *RSPO1* in the gonads of chicken embryos is estrogen-dependent. This study will provide a theoretical basis for studying the molecular mechanism of *RSPO1* in the development of chicken embryo ovary.

**Abstract:**

The mechanism of sex determination in chickens, especially the molecular mechanism of female ovarian development, has not yet been fully elucidated. Previous studies have shown that *RSPO1*, which is associated with ovarian development in mammals, might have a conserved role in chickens. In this study, we systematically investigated the spatiotemporal expression pattern of *RSPO1* in various tissues, especially gonads, of male and female chicken embryos using qPCR and Western blotting, and we explored its correlation with the expression of key genes in the estrogen pathway using drug treatment or gene overexpression in vivo and in vitro. Our results reveal that *RSPO1* was widely expressed in all examined tissues of chicken embryos, showing a female bias in gonadal tissues at both the mRNA and protein levels. Surprisingly, *RSPO1* was not differentially expressed between male and female gonadal cells with fadrozole-induced estrogen pathway blockades, and furthermore, estradiol-induced estrogen stimulation altered the expression of *RSPO1*. In addition, overexpression of *RSPO1* in gonadal cells induced the mRNA expression of its downstream target genes, *Wnt family member 4* (*WNT4*) and *Catenin beta 1* (*CTNNB1*), and that of *estrogen receptor α* (*ERα*), an estrogen pathway gene. In summary, this study provided new evidence for elucidating the role of *RSPO1* in ovarian development in poultry.

## 1. Introduction

In the poultry industry, the sex of animals is directly associated with economic traits [1,2]. For instance, in the egg-producing industry, a large number of male chickens are culled after hatching [3], whereas in the broiler industry, male chickens are preferred because of their higher growth rate and feed conversion ratio [4]. Therefore, controlling the poultry sex ratio is essential for improving the efficiency of poultry production. Overall, we have a preliminary understanding of male sex-determining genes in poultry, such as the *Z* chromosome gene *doublesex* and *mab-3 related transcription factor 1* (*DMRT1*), which is known to function as a testis determinant; however, the molecular mechanisms underlying female ovarian development have not been fully elucidated.

The *FOXL2/CYP19A1/ERα* signaling pathway, which is only activated in females (suppressed in males), is a well-known pathway that plays a key role in ovarian development. However, a recent study showed that although a male (ZZ) chicken with a single functional copy of *DMRT1* (generated by CRISPR-Cas9) developed ovaries in place of testes, which expressed *FOXL2/CYP19A1/ERα* as in the wild-type female, the male-to-female sex-reversed gonads could not further develop into functional ovaries [5,6]. This indicates that more factors may be involved in chicken ovary development. The *RSPO1/WNT4/β-catenin* pathway, which has been shown to be associated with female ovarian development in mammals, is likely to be one of the candidates [7,8]. *R-spondin 1* (*RSPO1*) is a member of the R-spondin family and encodes a protein containing a type 1 thrombospondin repeat sequence (TSR-1) [9], and it regulates the Wnt/β-catenin signaling pathway. Studies in humans have found that the loss-of-function mutation of *RSPO1* led to sex reversal in 46 XX women, indicating that it plays an important role in ovarian development [10]. In mice, *RSPO1* is expressed in embryonic gonads, and mutations in mouse *RSPO1* lead to masculinization, dysregulation of *WNT4* expression, and ectopic testosterone production in female mice [11]. Smith et al. found that *RSPO1* showed a conserved female-biased expression in gonads of chicken, mouse, and red-eared slider turtle [12]. Based on these findings, *RSPO1* is considered to be a regulator of vertebrate ovarian development [13]. However, the function of this gene, especially its interaction with the *FOXL2/CYP19A1/ERα* pathway, has not been well studied in poultry.

Therefore, we performed a systematic study to determine the spatiotemporal expression of *RSPO1* and to explore its relation with the *FOXL2/CYP19A1/ERα* pathway during chicken ovarian development. In this study, we determined the expression of *RSPO1* in both gonadal tissues and cells of E12 (12 days of incubation) male and female embryos by qPCR, detected the expression of the RSPO1 protein in gonads of E12 male and female embryos by Western blotting (WB), and explored the interaction between *RSPO1* and the estrogen pathway, another important pathway of ovarian development, at in vivo and in vitro levels using drug treatment and gene overexpression, respectively.

Our study showed that *RSPO1* exhibited a female-biased expression in chicken embryonic gonads, which was not inherently present in gonadal cells but was dependent on regulation by the estrogen pathway.

## 2. Materials and Methods

### 2.1. Egg Incubation and Sample Collection

Fertilized eggs of HY-LINE variety white chicken (domestic chicken) from a poultry farm (Yangzhou, China) were hatched in an intelligent incubator at 37.5 °C and 60% humidity. (The hens were raised according to the commercial management guides of Hy-line W-36, https://www.hyline.com/literature/W-36 (accessed on 11 May 2022).) For the spatiotemporal expression experiment, 100 eggs were incubated. For FAD and E2 injection groups and the control group, 60 eggs each were used. The blunt end was rotated upwards every 30 min until the required embryonic stage. On day 6.5 (E6.5), E9, E12.5, or E18.5, eggs were removed from the incubator, and the embryos were carefully dissected to expose the gonads. The gonadal tissues of E6.5, E9, E12.5, and E18.5 embryos were collected into 1.5 mL centrifuge tubes, quickly frozen in liquid nitrogen, and stored at −80 °C for subsequent RNA extraction. Tissues of liver, gonads, brain, spleen, intestine, kidney, stomach, heart, and muscle obtained from E12 (4 female and 4 male) were harvested for the organ-specific expression analysis. The gonads of E12 embryos were collected into 100 μL RIPA lysate (Cat No. C1053, APPLYGEN, Beijing, China) for protein extraction. For E6.5 and fadrozole (FAD)-injected chicken embryos, a small piece of tissue (wings or toes) was collected from each embryo to determine the genetic sex.

### 2.2. RNA Isolation, cDNA Synthesis, and Quantitative (Real-Time) Polymerase Chain Reaction (q-PCR)

Total RNA was extracted from gonads, cells, and other tissues using TRNzol Universal (TIANGEN, Beijing, China) reagent according to the manufacturer’s instructions. Briefly, gonads were collected into a 1.5 mL RNA enzyme-free centrifuge tube containing 1 mL of TRNzol reagent and homogenized with beads for 2 min. Then, 200 μL of chloroform was added to the lysate, the mix was shaken vigorously for 15 s, and then it was left at room temperature for 5 min. Subsequently, 1 mL of 75% ethanol was added to wash the total RNA, and then RNA was dried and dissolved with ddH_2_O. For E6.5 embryos, five pairs of gonads for each gender were pooled (4 pools generated for each sex). For E9, E12.5, and E18.5 embryos, total RNA was extracted from single pairs of gonads (*n* = 4 individuals for each sex). For sex reversal studies, total RNA was extracted from pairs of gonads from different groups (control male, control female, FAD-treated male, and FAD-treated female, *n* = 5 per group). First-strand cDNA was synthesized using a commercial kit (Cat No. R123-01, Vazyme Biotech Co., Ltd., Nanjing, China), according to the manufacturer’s instructions. Briefly, genomic DNA was removed by mixing 4 × gDNA wiper mix into total RNA and heating at 42 °C for 2 min, and then cDNA was amplified with 5 × qRT SuperMix II at 50 °C for 15 min, followed by a step at 85 °C for 2 min to inactivate the enzyme. Primers were designed according to NCBI, optimized for qPCR, and the most effective primer pair (efficiency > 95% and <105%) was selected. qRT-PCR was performed using an RT-PCR kit (Cat No. Q111-02/03, Vazyme Biotech Co., Ltd., Nanjing, China), and the level of mRNA expression was detected using the QuantStudio3 real-time PCR detection system (Thermo Fisher Scientific, Waltham, MA, USA). Relative expression was measured based on the expression of a housekeeping gene (*β-actin*) and was quantitatively analyzed using the 2^−ΔΔCq^ method. The primer sequences used for qRT-PCR are listed in Table 1.

### 2.3. Protein Extraction and Protein Imprinting

Total protein in gonads was extracted using the RIPA buffer according to the manufacturer’s instructions (Cat No. C1053, APPLYGEN, Beijing, China). Briefly, gonads were collected into a 1.5 mL centrifuge tube containing 300 μL RIPA buffer and 3 μL protease inhibitor and then blown to pieces by pipetting. After lysing on ice for 10 min, the lysate was centrifuged at 4 °C at 12,000× *g* for 10 min, and the supernatant, which contained the total protein, was moved to another tube until later use. For E12 embryos, total protein was extracted from the left gonads of each individual (*n* = 5 per sex). For FAD-treated embryos, total protein was extracted from a single pair of gonads in each treatment group (*n* = 3 per group). The relative level of RSPO1 protein in a single sample was estimated by analyzing the value of the target protein band in Western blots using the ImageJ V1.8.0.112 software (National Institutes of Health, Bethesda, MD, USA). The RSPO1 antibody (AF3474) was purchased from R&D Systems (Minneapolis, MN, USA), and staining with a tubulin protein antibody (#2144, CST, Peachtree, GA, USA) was used as an internal reference. The working concentrations of the RSPO1 and tubulin antibodies were 1:1500 and 1:1000, respectively.

### 2.4. Paraffin Section and Immunostaining

For histological analysis, the complex renal tissues were placed in 4% paraformaldehyde for 24 h. Next, the tissue shape was adjusted under a stereomicroscope. Subsequently, the tissue and corresponding labels were placed in a dehydration cassette, dehydrated with low to high ethanol concentrations, treated with xylene, and cleared for paraffin embedding. The embedded tissues were serially sliced to a thickness of 3 μm. Next, the slices were floated on a spreader of 40 °C warm water to spread the tissues. Then, the tissues were fished up with slides, baked in an oven at 60 °C, dried, and removed for storage at room temperature. Immunohistochemistry was performed as previously described [14]. Briefly, the slides were washed in PBS at 37 °C for 30 min and immersed in PBS containing 10% donkey serum, 1% BSA (Bovine Serum Albumin), and 0.3% Triton X-100 at 24 °C for 2 h. The slides were incubated with primary antibody overnight at 4 °C, washed in PBS containing 0.3% Triton X-100 before incubation, and then incubated with secondary antibody at room temperature for 2 h. Lastly, the slides were washed in PBS containing 0.3% Triton X-100, and the sections were treated with Hoechst solution (10 mg/mL) for 5 min to stain cell nuclei. The working concentrations of the antibodies used were as follows: goat anti-mouse R-Spondin 1 antigen affinity-purified polyclonal antibody (AF3474, R&D Systems, Minneapolis, MN, USA) was diluted 1:200 for immunostaining, and donkey anti-goat secondary antibody (GB21404, Servicebio, Wuhan, China) was diluted at 1:300 for immunofluorescence.

### 2.5. Construction of Overexpression Vector

According to the coding region of chicken *RSPO1* gene (NCBI ID: 419613, accession number: NM_001318444.2), the *RSPO1* overexpression vector was designed and synthesized by Shanghai Gima Gene Company (Shanghai, China) using a pcDNA3.1 vector. The empty vector was a pcDNA3.1 vector containing the CMV promoter. The same sequence of the RSPO1 open reading frame that was cloned and tested in vitro was PCR amplified from cDNA using forward primer 5′-GCTTGGTACCGAGCTCGGATCC-3′ and reverse primer 5′-TGCTGGATATCTGCAGAATTCCTATTGGGCAGGGCTGG-3′ with an included BamHI and EcoRI site. The resultant 783 bp fragment was subcloned into the NcoI and BamHI sites of pcDNA3.1 vector.

### 2.6. Acquisition, Culture, and Treatment of Chicken Embryo Gonad Cells

Chicken gonadal cells were isolated from the gonads of E12 HY-LINE variety white chicken embryos by 0.25% trypsin-EDTA (Gibco, Grand Island, NY, USA) digestion. First, the gonads were collected, washed three times with PBS (Solarbio, Beijing, China), and then trypsinized. Following the termination of digestion with 10% FBS-DMEM (Gibco, Hongkong, China), cells were washed with PBS, the supernatant was discarded, and cells were resuspended in 1 mL PBS. Centrifugation was performed at 700× *g* at 25 °C for 5 min, the supernatant was discarded, and 1 mL FBS-DMEM was added. A 70 µm nylon mesh was used to filter the resuspended cells into 50 mL conical tubes, to which complete medium was added, mixed well, and plated for culturing. Cells were seeded at 1 × 10^6^ cells/well in 12-well plates and cultured overnight in complete medium (DMEM medium with 10% fetal bovine serum, 1% penicillin streptomycin solution (100 IU/mL), and 10 μL EGF (PEPROTECH, Cranbury, NJ, USA) (20 ng/mL)). DMEM complete medium was prepared in advance. Cells reaching confluence were treated with β-estradiol. Before estradiol treatment, complete medium and working estradiol solutions were prepared at final concentrations of 0, 100, 150, 200, and 300 μmol/L. After treatment, cells were cultured for 24 h; five replicates were performed for each treatment. After washing with PBS, cells were collected and RNA was extracted using TRNzol Universal reagent. The transfection of the *RSPO1* overexpressing and empty plasmids in the control group was carried out according to the instructions for the jetPRIME^®^ in vitro DNA and siRNA transfection reagent (No. 101000046, Polyplus, Shanghai, China). The amount of overexpressing plasmid vectors and empty plasmid bodies was 0.8 μg per well, and cells were collected as described above.

### 2.7. Sexual Reversal and Genetic Sex Determination

To construct a model of sex reversal from female to male, 0.2 mg fadrozole (experimental group) or 100 μL PBS (control group) was injected into E2.5 eggs [15]. For the model of sex reversal from male to female, 17β-estradiol (E2; Sigma-Aldrich, St. Louis, MI, USA) was resuspended in 100% ethanol (10 mg/mL) and diluted to 1 mg/mL in sesame oil. Subsequently, 100 μL of 1 mg/mL solution (0.1 mg E2) (experimental group) or 10% ethanol sesame oil solution (control group) was injected into E2.5 eggs [16]. Briefly, a small hole (0.5 cm in diameter) was made in the blunt end of the egg, and then E2 or FAD was injected into the inner shell membrane above the chick embryo. A very small hole was then made in the membrane using sterilized tweezers, allowing the E2 and FAD fluid to diffuse into the embryo area. Then, the hole in the eggshell was sealed with a breathable medical tape, and the eggs were returned to incubation. The FAD injection group was incubated to E12, whereas the E2 injection group was incubated to E10.5. Follow-up experiments were conducted.

For genetic sex determination, a small piece of wing tissue was collected and used to extract DNA using a commercial kit (Cat No. DC102-01; Novozyme Biotechnology Co., Ltd., Nanjing, China). Subsequently, PCR was performed to amplify the CHD gene sequence located on both sex chromosomes. The CHD-forward/reverse primer sequences were as follows: F: AGTGCATTGCAGAAGCAATATT; R: GCCTCCTGTTTATTATAGAATTCAT. The female (ZW) had two bands at 506bp and 351bp, while the male (ZZ) had only one band at 506 bp [15].

### 2.8. Data Analysis

All statistical analyses were performed using the SPSS 22.0 software (IBM, Armon, NY, USA). Following qPCR, the relative gene expression was calculated using the 2^−ΔΔCq^ method. All experiments were repeated twice, and the data are expressed as the mean ± SEM. *t*-test was used for the significance analysis of data between two groups; Duncan test of one-way ANOVA analysis (SPSS 22.0) was used for data from E2 treatment of gonadal cells. Statistical significance was set at *p* < 0.05. GraphPad Prism V7 software (GraphPad, San Diego, CA, USA) was used for visualizing data (generating graphs and plots).

## 3. Results

### 3.1. RSPO1 Sequence Analysis

We compared the amino acid sequence of chicken RSPO1 with that of other species in the NCBI database using the DNAMAN V9.0 software (LynnonBiosoft, San Ramon, CA, USA) and constructed a phylogenetic tree using MEGA V11 software (Mega, Auckland, New Zealand). We determined the homology of the amino acid sequences of RSPO1 from 11 species of vertebrates, including chicken (*Gallus gallus*), Chinese turtle (*Pelodiscus sinensis*), human (*Homo sapiens*), mouse (*Mus musculus*), zebra fish (*Danio rerio*), zebra finch (*Taeniopygia guttata*), goat (*Capra hircus*), and common lizard *(Zootoca vivipara*). As shown in Figure 1A, the amino acid sequence of chicken RSPO1 had the highest similarity (96.1%) with that of the zebra finch RSPO1, lower similarities with that of human and mice RSPO1 (68.7% and 65%, respectively), and the lowest similarity (56.0%) with that of tilapia (*Oreochromis niloticus*) RSPO1. The RSPO1 proteins of these species include two FU (two cysteine-rich furin-like structural domains, Furin-like-1 and Furin-like-2) functional structural domains and one TSP1 (platelet response protein structural domain) functional structural domain. Interestingly, the adjacent FU structural domains were the most conserved major functional domains of the RSPO1 protein, and we assumed that RSPO1 has a relatively conserved function in these species. Phylogenetic tree analysis further confirmed that chicken RSPO1 had the highest homology to that of zebra finch (Figure 1B).

### 3.2. Temporal and Spatial Expression and Cellular Localization of RSPO1 in Chicken Embryos

We used *β-actin* as an internal reference to compare the levels of expression of *RSPO1* mRNA in the liver, gonad, brain, spleen, intestine, kidney, stomach, heart, and muscle tissues of E12 male and female chicken embryos. As shown in Figure 2A, *RSPO1* mRNA was expressed in all tissues of E12 male and female chicken embryos tested. Nonetheless, the expression was relatively high in the female gonads and spleen tissues of both sexes (more than twice as high as that in other tissues) and showed significant differences between males and females only in the gonads and kidneys; this difference was especially apparent in the gonads.

To further investigate the expression pattern of *RSPO1* in chicken embryonic gonads, we examined its mRNA expression in embryonic gonads of male and female embryos at different developmental stages. Our results are shown in Figure 2B. We found that *RSPO1* was expressed as early as E6.5, (early stage of gonad differentiation) in both male and female gonads, showing a low level of expression at both E6.5 and E9. However, from E12.5 to E18.5, its expression was dramatically increased in female gonads (approximately five-fold higher than that at E6.5 and E9), whereas it remained low in male gonads. Notably, the level of expression in females was consistently higher than that in males.

In addition, we examined the levels of expression of RSPO1 protein in E12 male and female gonads and its cellular localization in E12 female gonads using Western blotting and immunostaining, respectively. Consistent with the mRNA levels, the expression of RSPO1 was significantly higher in female gonads than that in male gonads (*p* < 0.01, Figure 2C). Figure 2D shows the mRNA expression levels of *RSPO1* in both the left and right gonads of E12 female chicken embryos, indicating that *RSPO1* expression in the degenerated right gonad is much lower than that in the left gonad. Interestingly, RSPO1 was mainly distributed in the cortical layer of E12 female gonads, whereas it showed weak or no expression in the medullary region (Figure 2E). In the male gonads, due to cortical degeneration, only weak expression of RSPO1 could be seen in the medulla (Figure 2E).

### 3.3. Effect of Estrogen Pathway on RSPO1 Expression in Chicken Embryonic Gonads

To investigate the effect of the estrogen pathway on the expression of *RSPO1* in the gonads of chicken embryos, we treated E2.5 chicken embryos with either the *AROM* (aromatase) inhibitor fadrozole (FAD) or estradiol to construct a sex reversal model. After identification of the genotypic sex by PCR analysis of the *CHD1* gene, we examined the mRNA and protein levels of RSPO1 using qPCR and Western blotting, respectively.

We observed that FAD treatment had no significant effect on the gonad morphology of ZZ-type chickens, whereas the gonads of ZW-type (female) chickens became masculinized, that is, the left gonad changed from enlarged to wrinkled compared with that in control females, whereas the right gonad did not degenerate (Figure 3A). Quantitative PCR analysis showed that the expression of the female marker gene *CYP19A1*, a key gene for estrogen synthesis, was significantly decreased in the gonads of females after FAD injection. Similarly, the expression of *FOXL2*, a key gene for ovarian development, was significantly reduced, whereas that of the key genes for masculinization, *DMRT1* and *AMH*, was upregulated in the gonads of masculinized females after FAD injection, demonstrating the effectiveness of the treatment (Figure 3B). Both the mRNA (Figure 3B) and protein (Figure 3C) expression of *RSPO1* were significantly reduced in the gonads of FAD-treated females compared with those in control females.

In contrast, β-estradiol injection did not result in any significant changes in gonadal morphology in ZW-type chickens, whereas the gonads of ZZ-type (male) chickens appeared feminized, that is, the left gonad changed from elongated to enlarged compared with that in control males, whereas the right gonad remained crinkled and degenerated (Figure 4A). Quantitative PCR analysis revealed that after β-estradiol injection, the expression of both *CYP19A1* and *FOXL2* was significantly upregulated, whereas that of *DMRT1* and *AMH* was significantly downregulated in feminized male gonads, demonstrating the effectiveness of the treatment. We also observed that the mRNA (Figure 4B) expression of *RSPO1* was significantly upregulated in β-estradiol treated male gonads compared with those in control males. These analyses suggested that the expression of *RSPO1* in chicken embryonic gonads is influenced by the estrogen pathway.

### 3.4. Nonindigenous Differences in RSPO1 Expression in Gonadal Cells and Effect of Estradiol Treatment

To further investigate the expression characteristics of *RSPO1* in chicken embryonic gonads and its relationship with the estrogen pathway, we isolated gonadal cells from E12 male and female embryos and examined the mRNA expression of *RSPO1*. Surprisingly, unlike gonadal tissues, there was no inherent difference in the expression of *RSPO1* between male and female gonadal cells. To exclude the effect of experimental manipulation, we simultaneously examined the expression of 3 genes of the estrogen pathway, *FOXL2*, *CYP19A1*, and *ERα*, and found that these genes were significantly differentially expressed between male and female gonadal cells (Figure 5A).

We thus hypothesized that the differences in the expression of *RSPO1* between male and female gonadal tissues were mediated by estrogen. To test this hypothesis, we treated male chicken embryonic gonadal cells with different concentrations of β-estradiol to investigate the effect of estrogen on the expression of *RSPO1*. Quantitative PCR analysis (Figure 5B) showed that the levels of mRNA expression of *RSPO1* in male gonadal cells treated with different concentrations of β-estradiol were significantly upregulated compared with those in the control group. Meantime, we also examined the effect of E2 treatment on the mRNA expression of *FOXL2*, *CYP19A1*, and *ERα*. Interestingly, the expression of *CYP19A1* and *ERα* could only be induced by a high concentration of E2 (200 µM, Figure 5C). In female gonadal cells, we found that the expression of *RSPO1* was also significantly upregulated after treatment with a 200 µM concentration of E2 (Figure 5D).

### 3.5. Effect of Overexpression of RSPO1 on Estrogen Pathway Genes in Male Embryonic Gonadal Cells

To further verify the relationship between the expression of *RSPO1* and the *FOXL2/CYP19A1/ERα* pathway, we constructed an *RSPO1*-overexpressing plasmid vector and transfected it into male chicken embryonic gonadal cells. We confirmed the transfection efficiency by qPCR, which showed that the level of expression of *RSPO1* mRNA was upregulated by 2300 times in the *RSPO1*-overexpressing group compared with that in the control group (Figure 6A).

Following overexpression of *RSPO1* in male gonad cells, we performed qPCR to detect the expression of its downstream target genes (*WNT4* and *CTNNB1*) and estrogen pathway genes (*FOXL2/CYP19A1/ERα*). We found that the expression of *WNT4* and *CTNNB1* was significantly increased in male chicken embryonic gonadal cells overexpressing *RSPO1* (Figure 6B). It is worth noting that the mRNA expression of *ERα* but not that of *FOXL2* or *CYP19A1* was upregulated after *RSPO1* overexpression in male chicken embryonic gonadal cells (Figure 6B).

## 4. Discussion

Although sexual differentiation in birds has been studied for decades, the specific molecular mechanisms remain unclear. Recent studies have revealed that *RSPO1* is closely associated with ovarian development in mammalian and scleractinian females and suggested that it might have a similar role in birds [12,17,18].

Our study aimed to investigate the spatiotemporal expression pattern of *RSPO1* in various tissues, especially the gonads, of male and female chicken embryos, and delineate the relationship between the expression of *RSPO1* and that of other key genes in the estrogen pathway, another important pathway of ovarian development, using drug treatment or gene overexpression both in vivo and in vitro. Our results confirm the female bias of expression of *RSPO1* in chicken embryonic gonad tissues; however, this bias was not inherent to gonadal cells but depended on estrogen stimulation. More specifically, we revealed a partial feedback loop regulation of the estrogen pathway by *RSPO1*, realized mainly through the modulation of the downstream target gene *ERα*. This study provided new evidence for the function and mechanism of *RSPO1* in chicken ovarian development.

A previous study found that *RSPO1* was expressed in a female-biased manner in vertebrate gonads, suggesting its functional conservation in ovarian development [12]. In addition, our comparison of the amino acid sequences of RSPO1 homologous proteins from different species supported the existence of similar structural RSPO1 domains among different species (Figure 1). However, the amino acid sequences of RSPO1 exhibited different degrees of variation among species. Previous studies have found differences in the function of RSPO1 among different species [19,20,21,22,23,24]. Therefore, although RSPO1 appears to have conserved functions in ovarian development among different species, its specific mechanisms of action or associated target genes might differ.

Smith et al. [12] examined the expression of *RSPO1* mRNA in E6.5, E8.5, E10.5, and E12 male and female embryonic gonads and found that *RSPO1* was expressed in a female-biased manner at all developmental stages. Specifically, *RSPO1* was consistently expressed at low levels in male gonads, whereas in females, its expression was gradually increased with gonadal development. This was consistent in part with the results of the present study (Figure 2B), namely, the female-biased expression of *RSPO1* and its consistently low expression in males. However, in our study, the expression of *RSPO1* in females did not gradually increase in the four periods examined (E6.5, E9, E12.5, and E18.5) but was relatively low at E6.5 (initiation of gonadal differentiation) and E9 (stage of gonadal differentiation), increased sharply at E12.5 (completion of gonadal differentiation, stage of ovarian development), and was maintained at that level until E18.5. This implied that *RSPO1* might be more strongly associated with ovarian development than gonadal differentiation. This notion was supported by the *RSPO1* expression in the left gonad of E12 females being much higher than that of the degenerating right gonad (Figure 2D), suggesting that the high expression of *RSPO1* in the left gonad at this stage was required for ovarian development. This is consistent with Smith’s whole in situ hybridization results, which showed *RSPO1* expression in the left and right gonads of female chicken embryos [12].

Furthermore, we verified the involvement of *RSPO1* in chicken ovarian development by its female-biased RSPO1 protein expression (Figure 2C). Our study confirmed that *RSPO1* was mainly distributed in the cortical layer (Figure 2E), which agreed with the results of Smith et al. [12]. Considering that the estrogen receptor ERα is also expressed in the ovarian cortex [25,26], it is likely that RSPO1 interacts with the *FOXL2/CYP19A1/ERα* axis, another important pathway for ovarian development in chicken embryos. Smith et al. [12] demonstrated that FAD-induced inhibition of estrogen synthesis reduced the expression of *RSPO1* in chicken embryos; we confirmed this in the present study (Figure 3C). However, it is not clear whether the effect of FAD treatment on *RSPO1* is mediated by estrogen, as FAD treatment apart from blocking estrogen synthesis also leads to changes in the expression of *AROM* and that of other genes in the pathway (such as *FOXL2*, Figure 3B). Therefore, we constructed a male-to-female sex reversal model by injecting estradiol into E2.5 chicken embryos and directly confirmed that estradiol stimulation induced the expression of *RSPO1* in chicken embryonic gonadal tissues (Figure 4).

These findings confirm that the estrogen pathway regulates the expression of *RSPO1* in chicken embryonic gonadal tissues, causing its female-biased high expression. Cheng et al. [27] treated Chinese soft-shelled turtle embryos before sex differentiation using different concentrations of estradiol and found that it resulted in significantly increased expression of *FOXL2*, *CYP19A1*, *WNT4*, and *RSPO1*. In addition, overexpression of the key enzyme CYP19A1, responsible for estrogen synthesis in male gonads, can upregulate the expression of *RSPO1* [28]. However, whether this female-biased expression of *RSPO1* was estrogen-dependent remains unclear. To answer this, we isolated gonadal somatic cells from differentiated male and female gonads (E12) and examined the expression of *RSPO1* mRNA and that of other genes after 2 d of culture. Of note, Smith et al. [12] demonstrated that *RSPO1* was mainly expressed in mouse gonadal somatic cells. However, the expression of *RSPO1* in chicken embryonic gonadal cells was not detected. Here we found that, unlike *FOXL2/CYP19A1/ERα*, there was no cell-intrinsic difference in the expression of *RSPO1* between male and female gonadal cells; this pattern was different from the female-biased expression of *RSPO1* in gonadal tissues. We hypothesized that the high expression of *RSPO1* in female gonads might be dependent on estrogen stimulation, as the in vitro culture estrogen synthesis was inhibited despite the high expression of *FOXL2/CYP19A1/ERα* in female gonadal cells, probably due to lack of required substrates. Therefore, we treated gonadal cells with estradiol and found that stimulation with estradiol significantly upregulated the expression of *RSPO1*, indicating that the female-biased expression of *RSPO1* was estrogen-dependent. The expression of *ERα*, *CYP19A1*, and *FOXL2*, key genes for female pathway development, were also upregulated in estrogen-treated male gonadal cells, further demonstrating the effectiveness of estrogen treatment. Cai et al. [29] also reported enhanced expression of *RSPO1* during estrus, a phase with high estrogenic signaling activity, suggesting that *RSPO1* is a hormone-mediated local factor whose expression could be upregulated by estrogen and progesterone [30].

Regarding the mechanism of function of *RSPO1*, studies in mammals have suggested that *WNT4* and *β-catenin* might be downstream target genes [10,11,31,32], with *RSPO1* activating the β-catenin pathway together with *WNT4* to participate in ovarian development [11]. In the present study, we found a similar regulatory relationship in chickens. Overexpression of *RSPO1* in chicken embryonic gonadal cells induced the mRNA expression of *WNT4* and *CTNNB1* (Figure 6). Interestingly, we identified a partial feedback loop regulation of the *FOXL2/CYP19A1/ERα* pathway by *RSPO1*, as the levels of *ERα* mRNA were upregulated after overexpression of *RSPO1*, whereas the expression of the other 2 genes, *FOXL2* and *CYP19A1*, did not show significant changes, nor did the expression of the key gene for masculinization, *DMRT1*. This indicated that despite the feedback loop regulation of the downstream factors of the *FOXL2/CYP19A1/ERα* pathway to a certain extent, *RSPO1* does not directly affect the upstream sex-determining genes. In line with our results, Geng et al. [33] also found that *RSPO1* promoted *ERα* expression in mouse mammary duct luminal cells. This is consistent with our finding that the expression of *ERα* was significantly upregulated after overexpression of *RSPO1* in chicken embryonic male gonadal cells. However, Zhang et al. [24] observed a significant decrease in the expression of *FOXL2* after knocking down the expression of *RSPO1* in the gonads of female Chinese soft-shelled turtles in vivo, which demonstrates the feedback regulation of *RSPO1* on the *FOXL2/CYP19A1/ERα* pathway and also highlights the differences in the degree of this feedback among different species. The above studies demonstrate the importance of *RSPO1* in ovarian development and suggest that there is indeed an interaction between *RSPO1* and the estrogen pathway, but the exact mechanisms need to be further investigated.

## 5. Conclusions

In conclusion, although both the female-biased expression of *RSPO1* in chicken embryonic gonads and the effect of interfering with estrogen synthesis on the expression of *RSPO1* have been previously reported, the present study further clarified the spatiotemporal expression pattern of *RSPO1* in chicken embryonic tissues and directly confirmed the regulatory role of estrogen in the expression of *RSPO1* in gonadal tissues and cells. More importantly, this study revealed that the differential expression of *RSPO1* in male and female gonads is not cell-intrinsic but dependent on estrogen induction and controlled through a partial feedback loop regulation of the *FOXL2/CYP19A1/ERα* pathway by *RSPO1*. However, the role of *RSPO1* in avian gonadal development, especially in late ovarian development, needs further validation in vivo. To this end, the construction of lentiviral-based interference and overexpressing vectors for the transfection of chicken embryos or the construction of transgenic chickens by knocking down *RSPO1* using CRISPR/Cas9 would be useful in future studies.

## Figures and Tables

**Figure 1 animals-13-02240-f001:**
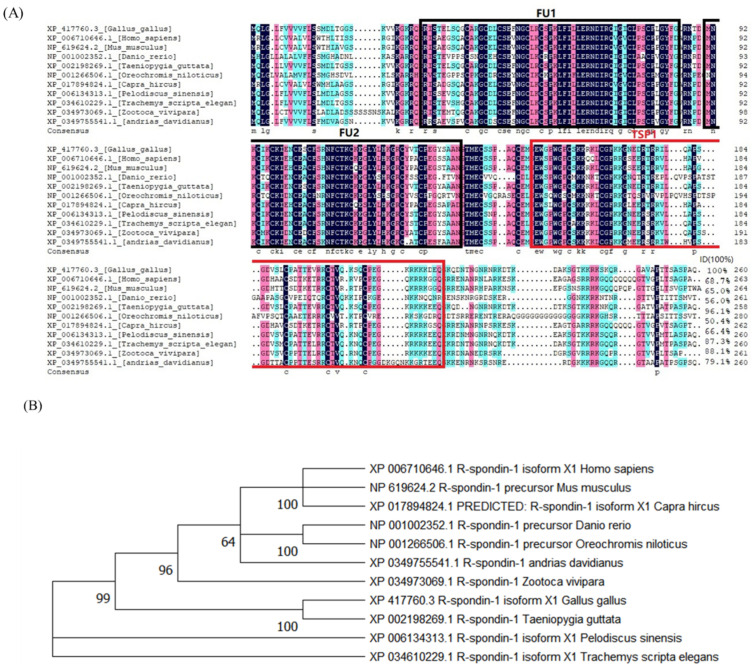
Sequence analysis of Chicken RSPO1. (**A**) Alignment of amino acid sequences of chicken RSPO1 protein with other species. Shaded areas indicate homologous amino acids, black indicates 100% homology, pink indicates >75% homology, light blue indicates >50% homology. FU (furin-like): two cysteine-rich furin-like structural domains (marked with black box); TSP1: platelet response protein structural domain (marked with red box); LC (low complexity): low-complexity region. (**B**) Phylogenetic tree analysis of RSPO1 in chicken and other species of vertebrates.

**Figure 2 animals-13-02240-f002:**
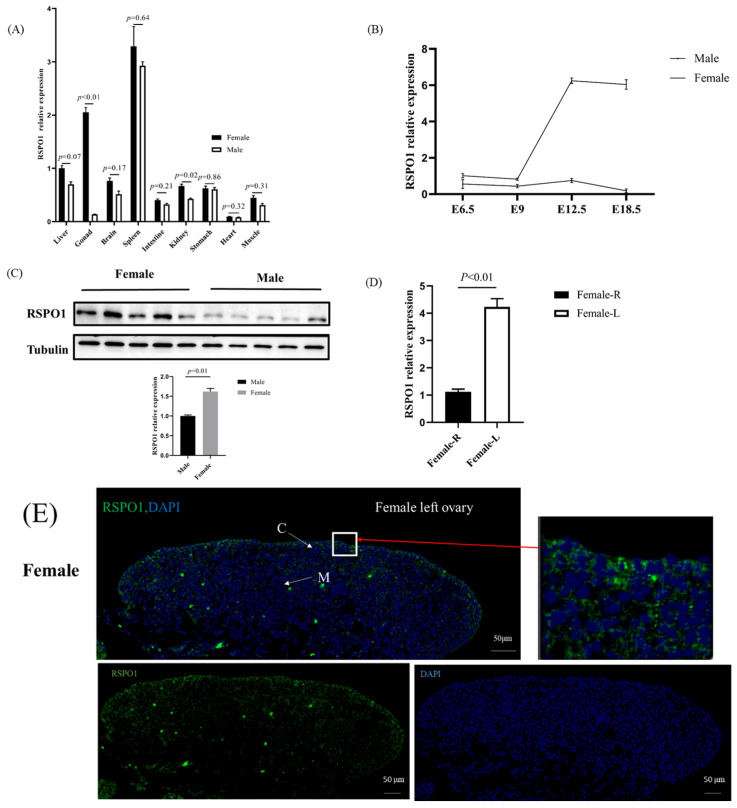
Spatiotemporal expression of chicken *RSPO1*. (**A**) Relative mRNA expression of *RSPO1* in tissues of E12 (HH 38) male and female chicken embryos. *n* = 4. (**B**) Relative mRNA expression pattern of *RSPO1* at different stages of chicken embryos, E12 =12 days of incubation. Beta-actin was used as a reference gene. *n* = 4. (**C**) Relative protein expression of RSPO1 in male and female gonads of E12 chicken embryos. Data are shown as mean ± SEM. Tubulin was used as a reference gene. *n* = 5. (**D**) *RSPO1* mRNA expression levels in left and right gonads of E12 chick embryos. (**E**) Localization of RSPO1 protein expression in the ovaries and testes of E12 chick embryos. Scale bars = 50 μm. M: medulla. C: cortex.

**Figure 3 animals-13-02240-f003:**
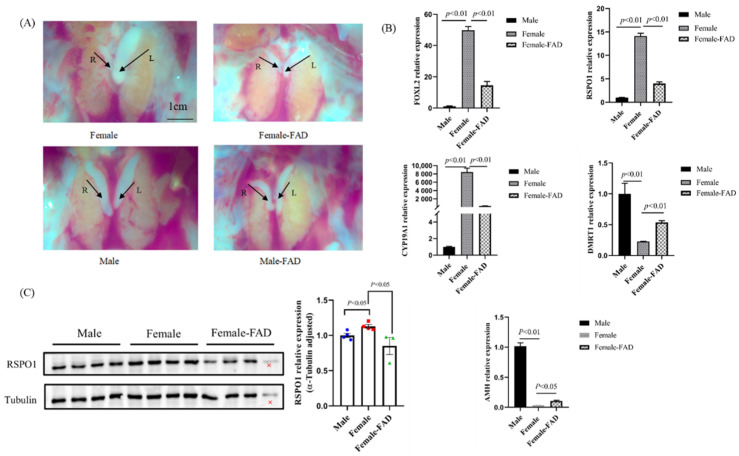
Effect of FAD treatment on the expression of *RSPO1* in E12 chicken embryonic gonads. (**A**) Anatomy of male and female gonads of FAD-treated group and control group. L = left gonad, R = right gonad. Scale bars = 1cm. (**B**) Effect of FAD treatment on the mRNA expression of *RSPO1* and some other genes in chicken embryonic gonads. Beta-actin was used as a reference gene. *n* = 4. (**C**) Effect of FAD treatment on the protein expression of RSPO1 in chicken embryonic gonads (the wrong symbol X indicates a false sample loading whose result was not included in the analysis). *n* = 4. Tubulin was used as a reference gene, and data are shown as mean ± SEM. FAD: fadrozole. FAD treatment was performed on E2.5 (embryonic day 2.5). E12 = embryonic day 12.

**Figure 4 animals-13-02240-f004:**
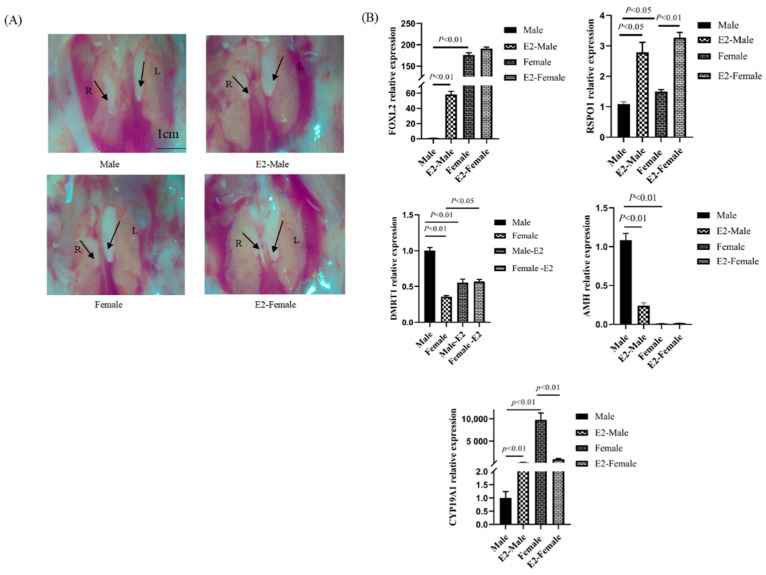
Effect of E2 treatment on the expression of *RSPO1* in E10.5 chicken embryonic gonads. (**A**) Anatomy of male and female gonads of E2-treated group and control group. L = left gonad, R = right gonad. Scale bars = 1cm. (**B**) Effect of E2 treatment on the mRNA expression of *RSPO1* and some other genes in chicken embryonic gonads. Beta-actin was used as a reference gene. *n* = 4. E2: estradiol. E2 treatment was performed on E2.5 (embryonic day 2.5). E10.5 = embryonic day 10.5.

**Figure 5 animals-13-02240-f005:**
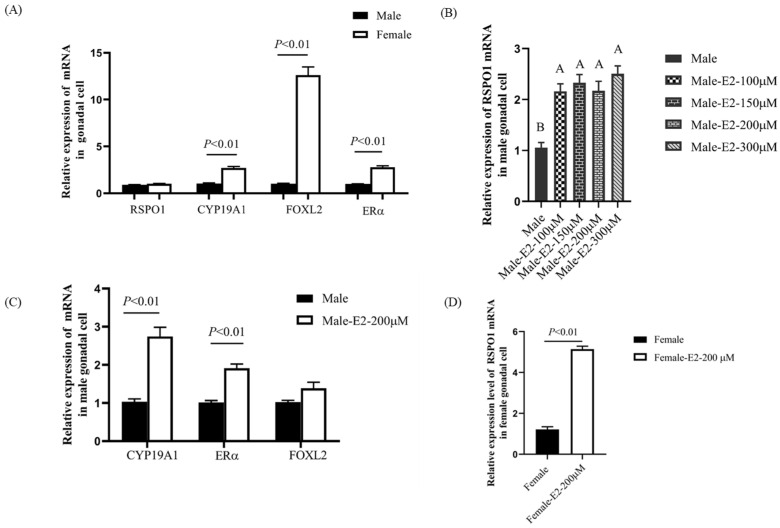
Expression of *RSPO1* and estrogen pathway genes in gonadal cells and their response to E2. (**A**) Intrinsic expression of *RSPO1* and estrogen pathway genes in gonadal cells. *n* = 6. (**B**) Effect of E2 treatment on *RSPO1* mRNA expression in male gonadal cells. *n* = 5. male-E2-100/150/200/300 μM: male gonadal cells treated with 100/150/200/300 μM estradiol. Bars with different capital letters (A, B) are significantly different (*p* < 0.05). (**C**) Effect of E2 treatment on mRNA expression of estrogen pathway genes in male gonadal cells (*n* = 4). (**D**) Effect of E2 treatment on mRNA expression of *RSPO1* in female gonadal cells (*n* = 6). *Beta-actin* is used as a reference gene, and data are shown as mean ± SEM. E2: estradiol.

**Figure 6 animals-13-02240-f006:**
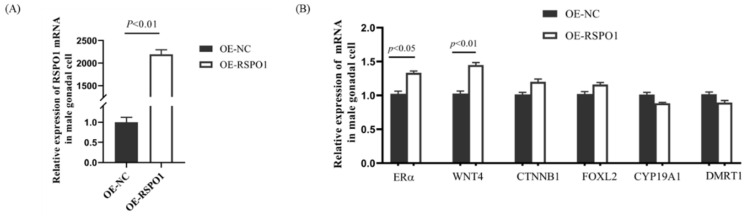
Effects of *RSPO1* overexpression on mRNA expression of downstream genes and estrogen pathway genes in gonadal cells. (**A**) *RSPO1* mRNA expression in overexpression group and control group. *n* = 6. (**B**) Effect of *RSPO1* overexpression on mRNA expression of *WNT4*, *CTNNB1*, and estrogen pathway genes in male gonadal cells. *n* = 6. *Beta-actin* was used as a reference gene, and data are shown as mean ± SEM.

**Table 1 animals-13-02240-t001:** Sequence information of PCR primers.

Gene Name	Primer (5′–3′)	Gene ID
*RSPO1*	F:GGCAGCAAAGTGGTGAAG	NM_001318444.2
	R:GACAGGATGGGAGGCAGA	
*CYP19A1*	F:TGTTCCATCACGCTATTT	NM_001001761.4
	R:GATTCTTGTTTGGGCTTC	
*DMRT1*	F:TTCGCGTTGAGTGCCTCGAC	NM_001101831.3
	R:GAGGACACTGTGAGCCGTTC	
*FOXL2*	F:ACATGTTCGAGAAGGGCAAC	NM_001012612.1
	R:TGTTCATGAAGGTGGACTGC	
*ERα*	F:CTACTGGCTACTGCTGCTCG	NM_205183.2
	R:AGGTGCTCCATTCCTTTGT	
*AMH*	F:GAAGCATTTTGGGGACTGG	NM_205030.2
	R:GGGTGGTAGCAGAAGCTGAG	
*CTNNB1*	F:GCCGGGCACTATTTCTCCTC	NM_205081.3
	R:CCGGGCGAGAGGCTTAAAAT	
*β-actin*	F:TATGTGCAAGGCCGGTTTC	NM_205518.2
	R:TGTCTTTCTGGCCCATACCAA	

## Data Availability

The data presented in this study are available on request from the corresponding author for scientific purposes.

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
