# Peer review of "Female-Biased Expression of R-spondin 1 in Chicken Embryonic Gonads Is Estrogen-Dependent"

_animals, 2023, doi:10.3390/ani13132240_

Round 1

Reviewer 1 Report

The article titled: Female biased expression of RSPO1 in chicken embryonic gonads is estrogen dependent, and written by Mingde Zheng, et all. was read and evaluated carefully.

When reading the article, it seems difficult to understand, even for connoisseurs. The authors use a fairly simple introduction - more information should be provided about the role of gene expression knowledge and identification methods.

The methodology is correctly described and uses current technology

The results should be compared with several other researches.

The entire manuscript needs proofreading and revision, there are many small mistakes in writing and spelling

 Moderate editing of English language required

Author Response

Dear reviewers:
Thank you and the reviewers very much for the opportunity to improve our manuscript entitled “Female biased expression of RSPO1 in chicken embryonic gonads is estrogen dependent” (ID: animals-2429402). We have carefully considered all the comments and suggestions and made the corresponding revisions (Any changes to the manuscript have been marked by “Track Changes ” ). And the responses to the reviewer’s comments are as flows.

Thank you and best regards

Jun Zhang

Responses to the reviewers’ comments:

Comment 1:When reading the article, it seems difficult to understand, even for connoisseurs. The authors use a fairly simple introduction - more information should be provided about the role of gene expression knowledge and identification methods.

Response: Thanks for your suggestion. We have added some more information on the expression of Rspo1 and its implication to make the introduction part more informative (Line 57-66).

Comment 2: The results should be compared with several other researches.

Response: Thanks for your suggestion, and we have compared our results with some other studies in the discussion section (Line 485-487, 513-514, 547-549).

Comment 3: The entire manuscript needs proofreading and revision, there are many small mistakes in writing and spelling

Response: Sorry for the mistakes. We have revised the writing and spelling errors in the manuscript (marked with track changes).

Reviewer 2 Report

There are formatting issues throughout this manuscript; including words that run together (e.g., L 58, L 194); words that are broken apart (e.g., L37, L39, L 41, L 44,); and lack of spaces between periods, numbers, commas, etc., 

The introduction does not effectively set the stage for cause and effect and the rationale for only looking at RSPO1 and why it is relevant and important. 

Data does not support the statement "RSPO1 is female-biased" when you find it in all embryos and there is no differential expression between males and females? 

L56-63 Does not set the stage for cause and effect. No hypothesis.

L64-66: Do not agree with this based on previous comments and this does not tell us what it means. 

Why is RSPO 1 important?  Never tell us why.

Discussion--vague.  Does not tell us what your findings mean.  Why is the RSPO1 gene important? why did you target it?  The discussion needs to be rewritten to inform the reader what your findings mean and why they are important.     

English/Grammar is okay, minor problems.  The formatting is the major problem with this manuscript.  

Author Response

"Please see the attachment

Reviewer 3 Report

In the article entitled “Female biased expression of RSPO1 in chicken embryonic gonads is estrogen dependent” Zheng et al. evaluated the role of RSPO1 in ovarian development in chicken. They used quantitative gene expression profiles of RSPO1 in various tissues including gonads.

Current research will definitely improve our understanding of the regulatory roles of estrogen on the expression of RSPO1 genes –probably controlled through feedback loop regulation of other pathway genes (FOXL2/CYP19A1/Erα) by RSPO1.

The experiment is well designed and the manuscript is written very concisely conveying the message clearly to the readers.

Minor comment#

The manuscript must be proofread for typo errors such as spelling, spacing, and rules of scientific names both in the manuscript and on phylogenetic figure

The manuscript must be proofread for typo errors such as spelling

Reviewer 4 Report

In general, the paper is well-written. Relevant and current literature in the study was cited. However, the authors need to address the following major concerns:

·       The introduction should be beefed up to give a clearer picture of concepts raised in the study as reported by other researchers who might have done similar work.

·       The authors should have highlighted the husbandry system in raising the chickens used in this study, including feeds and feeding. It is important to have given a brief on the composition of the experimental diet and a table to show this. The level of nutrition must have had a dramatic effect on the expression of genes and will afford other interested researchers to conduct this or a similar study.

·       The authors have written the same word (upregulate and up-regulate) with and without a hyphen; both are acceptable, but it’s best to be consistent by adopting one of them.

Furthermore, the following specific comments should equally be addressed by the authors:

Title: The title is well written, but RSPOI should be written in full.

Simple summary

Line 8 RSPO1 should be defined by writing it in full, and the RSPO1 should be put in parenthesis and subsequently left in an abbreviated form.

Line 14 And is not the best choice here. Consider replacing it with a more suitable word. Suggestion: Furthermore or moreover.

Abstract

This is well-written but could still be improved upon. Most of your readers will only browse through the abstract and may not read other abstracts of the paper. This section should be improved upon.

Line 30 WNT4, CTNNB1, and ERα are at first mentioned here, they should be defined by writing them in full and put in parentheses, subsequently left in an abbreviated form. This should apply to other abbreviations used in subsequent parts of the manuscript.

Line 32 Separate ‘of’ from ‘RSPO1’.

Introduction

Line 42 ‘as’ should be separated from the asdoublesex. Suggestion: as doublesex.

Lines 64-66 This aspect looks more like a significant finding from the study and should be incorporated in the conclusion rather than being part of the introduction.

Materials and methods

Line 69 It is worthwhile to state how many of the Fertilized eggs of HY-LINE variety white chicken from a poultry farm were hatched in an intelligent incubator.

Line 84 The authors didn’t state or elaborate on how the total RNA used in this study was extracted. There is a need to state the source of the kits used and how the extraction was carried out in two to three sentences. The authors stated the commercial kit that was used for the synthesis of first-strand cDNA, it is essential to state in a few sentences the procedure for their readers to follow and possibly replicate the study.

Line 116 Total protein in gonads was extracted using the RIPA buffer according to the manufacturer’s instructions (CatNo.C1053, APPLYGEN, China). Please summarise the process in a few sentences.

Results

Lines 214-216 The scientific words in parentheses should be italicised.

RSPO1 in this paragraph and elsewhere should be italicised. Check other genes in the manuscript to ensure uniformity in the presentation.

Discussion

Line 388 evidencefor, these words should be separated. Suggestion: evidence for.

Line 411 Thisnotion, these words should be separated. Suggestion: This notion.

Line 414 forits, these words should be separated. Suggestion: for its.

Line 418 et al[7] is incorrectly punctuated. Think about changing the punctuation.

Line 422 embryos;we, are improperly spaced; consider adding a space.

Line 436 [24].However, are improperly spaced; consider adding a space.

.

Round 2

Reviewer 2 Report

the authors adequately addressed my concerns 

Author Response

Thanks to the reviewer for your recognition and your kind help during the revision of our manuscript.